

# Assessment of nocturnal Aerosol Optical Depth from lunar photometry at Izaña high mountain Observatory

África Barreto[1,2,5], Roberto Román[3,4], Emilio Cuevas[1], Alberto J. Berjón[5], A. Fernando Almansa[1,2,5], Carlos Toledano[5], Ramiro González[5], Yballa Hernández[1], Luc Blarel[6], Philippe Goloub[6], and Margarita Yela[7]

[1]Izaña Atmospheric Research Center, Meteorological State Agency of Spain (AEMET), Spain
[2]Cimel Electronique, Paris, France
[3]Department of Applied Physics, University of Granada, Granada, Spain
[4]Andalusian Institute for Earth System Research, IISTA-CEAMA, University of Granada, Junta de Andalucía, Granada, Spain
[5]Grupo de Óptica Atmosférica, Universidad de Valladolid, Valladolid, Spain
[6]Laboratoire d'Optique Atmosphérique (LOA), Université de Lille, Villeneuve d'Ascq, France
[7]Instrumentation and Atmospheric Research Department, National Institute for Aerospace Technology (INTA), Madrid, Spain

*Correspondence to:* Emilio Cuevas
(ecuevasa@aemet.es)

**Abstract.** This work is a first approach to correct the systematic errors observed in the aerosol optical depth (AOD) retrieved at night-time using lunar photometry and calibration techniques dependent on the lunar irradiance model. To this end, nocturnal AOD measurements were performed in 2014 using the CE318-T master Sun-sky-lunar photometer (Lunar-Langley calibrated) at Izaña high mountain Observatory. This information has been restricted to 59 nights characterized as clean and stable according-

ing to lidar vertical profiles. A phase angle dependence as well as an asymmetry within the Moon's cycle of the ROLO model could be deduced from the comparison in this 59-nights period of the CE318-T calibration performed by means of the Lunar-Langley and the calibration performed every single night by means of the common Langley technique. Nocturnal AOD has also been compared in the same period with a reference AOD based on daylight AOD extracted from the AERONET network at the same station. Considering stable conditions, the difference $\Delta AOD_{fit}$, between AOD from lunar observations and the

linearly interpolated AOD (the reference) from daylight data, has been calculated. The results show that $\Delta AOD_{fit}$ values are strongly affected by Moon phase and zenith angles. This dependency has been parameterized using an empirical model with two independent variables (Moon phase and zenith angles) in order to correct the AOD for these residual dependencies. The correction of this parameterized dependency has been checked at four stations with quite different environmental conditions (Izaña, Lille, Carpentras and Dakar) showing a significant reduction of the AOD dependence on phase and zenith angles, and

an improved agreement with daylight reference data. After the correction, absolute AOD differences for day-night-day clean and stable transitions remain below 0.01 for all wavelengths.



# 1 INTRODUCTION

Aerosols can significantly influence the climate in several ways: through aerosol-radiation and aerosol-cloud-precipitation interactions (Foster et al., 2007; IPCC, 2013). This fact has motivated notable efforts in atmospheric sciences envisaged to increase the understanding of the role played by aerosols in the global climate balance.

Passive and active remote sensing techniques have recently experienced a great advance providing aerosol concentration and properties with high precision and spatial coverage. Aerosol optical depth (AOD) is a valuable parameter accounting for aerosol load in the atmosphere because it is a measure of the extinction of the solar beam by absorption and scattering processes caused by aerosols. Among the passive techniques, it is worth mentioning the use of Sun photometry to retrieve columnar aerosol optical and microphysical properties, which provides useful information to characterize aerosols with an excellent

spatial coverage but with the lack of vertical resolution (Holben et al., 1998; Eck et al., 1999; Holben et al., 2001; Eck et al., 2009, 2010). A good example of the spatial extent of Sun photometry techniques is the widespread ground-based AErosol RObotic NETwork (AERONET) (Holben et al., 1998) and its federated networks, including hundreds of stations globally distributed. Other networks are SKYNET, China Aerosol Remote Sensing Network (CARSNET) and Global Atmospheric Watch–Precision Filter Radiometer (GAW-PFR). However, aerosols at night-time have been studied to a much lesser extent

(Barreto et al., 2013a, b; Baibakov et al., 2015). There is a growing interest in studying the diurnal dynamics and evolution of atmospheric aerosols (Pérez-Ramírez et al., 2012a), as well as understanding the nucleating role of aerosols and their net radiative effects (Baibakov et al., 2015). Therefore, new technological developments try to fill the night-period gaps in AOD time series. As Baibakov et al. (2015) pointed out, star and Moon photometry have arisen as plausible solutions to this problem. Star photometer technique (Leitener et al., 1995; Pérez-Ramírez et al., 2015; Baibakov et al., 2015) has been revealed

as a useful tool to infer aerosol information during night-period. However, infrastructure and logistic constraints still represent an important limitation for the operational use of stellar measurements, especially for global networks such as AERONET. Alternatively, Moon photometry is a technique that can be implemented more easily, and at a lower cost, in an operational way (Barreto et al., 2016). Nevertheless, Moon photometry technique is still affected by notable limitations. Despite the Moon is our nearest celestial neighbor, our knowledge about its spectral irradiance is far from being as precise as the spectra from the

Sun or bright stars like Vega (Cramer et al., 2013).

    The main important obstacle in Moon photometry is the fact that the Moon is a variable reflector of sunlight and, as a result, it is a highly variable source of visible light (Miller et al., 2012). In addition, lunar orbit is rather complex, highly elliptical compared to the Earth's orbit around the Sun, and the Moon's brightness is highly nonlinear in its cycle (Miller et al., 2012). All these problems mean that the development of the Moon photometry is not an easy task. Esposito et al. (1998) and Berkoff et al.

(2011) made pioneering works in lunar photometry. Berkoff et al. (2011) used a modified Sun photometer to obtain night-period AOD measurements using moonlight. These authors used a lunar irradiance model to account for the continuous changes in the Moon's illumination over the Moon cycle. Barreto et al. (2013a, b) presented a new photometer prototype (CE318-U), similar to the CE318-AERONET reference instrument, which included sensors with a high degree of amplification to collect visible light at night-time, allowing nocturnal aerosols and water vapor monitoring. Measurements using this new generation of lunar



photometers cover $\sim$ 50% of the Moon cycle, significantly extending the continuity of the existing aerosols observations, and specifically during the polar night. Recently, Barreto et al. (2016) presented the new photometer CE318-T which combines the features of the extensively used by the Cimel Sun photometer, standard model in AERONET network, with the lunar photometer prototype previously presented in Barreto et al. (2013a, b). The higher precision of this new instrument compared to the previous versions of Sun and Moon photometers and its ability to monitor atmospheric aerosols in a diurnal cycle, have made it a suitable instrument to replace the CE318-AERONET reference instrument.

As many authors have stated (Berkoff et al., 2011; Barreto et al., 2013a, b, 2016), a precise Moon irradiance model is mandatory in Moon photometry to take the continuous change of Moon's brightness over the cycle into account. In this respect, RObotic Lunar Observatory (ROLO) model, developed by Kieffer and Stone (2005), is the most careful radiometric study on the Moon's brightness to date (Cramer et al., 2013). It is an empirically based model made at United States Geological Survey (USGS) Robotic lunar Observatory in Flagstaff, Arizona, using ground-based photometric observations spatially resolved from 1996 to 2003, capturing the details of lunar maria and lunar terrae . This model is widely used in the literature providing precise knowledge of the Moon's spectral irradiance (Eplee et al., 2011, 2012; Lacherade et al., 2013; Viticchié et al., 2013), taking advantage of the high stability of the Moon's surface properties to use this celestial body for space-borne calibration of visible-band sensors. The ROLO model has recently emerged as a unique tool for Moon photometry (Berkoff et al., 2011; Barreto et al., 2013a, b, 2016), and is an essential part of the calibration process. Although this model provides precise information about the change of Moon's irradiance with the phase angle ($g$) and lunar librations, as well as about the non-uniform surface albedo and the non-lambertian Moon's reflectance, small systematic effects have been found in this model. Lacherade et al. (2013) and Viticchié et al. (2013) found a small phase angle dependence of the ROLO calibration using the Pleiades Orbital Lunar Observations (POLO) and Meteosat Second Generation (MSG) Spinning Enhanced Visible and Infrared Imager (SEVIRI) solar bands. Cramer et al. (2013) developed a novel apparatus to accurately measure the lunar spectral irradiance with the aim of estimating these systematic effects in the ROLO model. Barreto et al. (2016) used the CE318-T and the ROLO model to retrieve AOD at day and night-time in Izaña, a high altitude observatory located at Tenerife (The Canary Islands, Spain). These authors observed an important dependence of the AOD uncertainty with phase angle and also a faint nocturnal cycle in AOD, indicating a possible dependence of AOD uncertainty on the Moon's zenith ($\theta$) and phase angles. As these authors stated, the reason for these discrepancies remain unclear, although it is likely to be due to a sum of causes, such as inaccurate instrument calibration, possible systematic errors in the ROLO model, and uncertainties in night-time AOD calculation.

This work is based on all of the previous results to improve the AOD retrieval at night-time by selecting a set of clean and stable night-time conditions at Izaña in which day-time AOD data could be considered a good proxy for nocturnal AOD. Clean and stable conditions of days used in this study have been ensured using AERONET daytime data at the station and Micropulse lidar version 3 (MPL-3) atmospheric vertical profiles extracted from a nearby coastal station. The main aim of this study is to identify the errors sources, thereby trying to fix experimentally some of the problems currently affecting Moon photometry.

Section 2 describes the experiment site, instruments and methods used in this study. A description of the methodology developed to improve nocturnal AOD measurements and the corresponding validation performed at Izaña, as well as in other complementary stations, is presented in section 3. Finally conclusions are shown in section 4.





## 2 MEASUREMENT SITE

### 2.1 Test site

Nocturnal measurements have been carried out at Izaña Global Atmosphere Watch (GAW) Observatory (http://izana.aemet.es), managed by the Izaña Atmospheric Research Center (IARC) from the State Meteorological Agency of Spain (AEMET).

The Izaña Observatory is a testbed for aerosols and water vapor remote sensing instruments of the World Meteorological Organization (WMO) Commission for Instruments and Methods of Observations (CIMO). It is a high mountain station (2373m a.s.l.) located in Tenerife (The Canary islands, Spain) at 28° 18' N, 16° 29'W. The main features of this station have been extensively described by Rodríguez et al. (2011); Cuevas et al. (2013); Guirado (2014) and Cuevas et al. (2015).

The station is characterized by NW subsiding air from the descending branch of the Hadley-cell, resulting in a strong

temperature inversion normally located below the altitude of the station (800 to 1500 m a.s l.). This structure usually separates the humid layer, potentially laden with some anthropogenic pollution from lower parts of the island, from the dry and clean troposphere above. Environmental conditions at Izaña make the site quite suitable for aerosol sensors calibrations because the wide range of AOD values: from AOD at 500 nm ($AOD_{500}$) below 0.01 under background almost-Rayleigh conditions to $AOD_{500} > 0.15$ under Saharan dust intrusions. Around 85% of the days present quite stable and low $AOD_{500}$ values, below

0.15 (Guirado, 2014). Pristine conditions make Izaña a suitable place to calibrate photometers using the Langley method.

### 2.2 Instruments

#### 2.2.1 CE318-T photometer

The new Sun-sky-lunar multiband photometer (CE318-T) has recently been presented in Barreto et al. (2016) as an advanced system which combines the features of the Sun photometer CE318-N, extensively used as a reference instrument in AERONET

network (Holben et al., 1998), with the lunar photometer prototype CE318-U presented in Barreto et al. (2013a, b). The new CE318-T photometer is capable of measuring Sun, Moon and sky radiances at an approximate field of view of 1.29° at eight nominal wavelengths of 1020, 937, 870, 675, 500, 440, 380 and 340 nm, using a silicon photodiode detector, as well as two additional channels at 1020 nm and 1640 nm using an InGaAs detector. Silicon 1020 nm channel has been temperature corrected following the methodology presented in Holben et al. (1998). The UV channels do not allow an accurate AOD

retrieval at night due to the low lunar signal in this wavelength range. The CE318-T master used in the present study has been calibrated by means of the Lunar-Langley calibration method presented in Barreto et al. (2013a). Cloud-screening of night-time AOD data has been performed by visual inspection and using the triplet criterion presented in Barreto et al. (2016).

#### 2.2.2 MPL-3 Lidar

Vertical range corrected signal profiles from the MPL-3 Lidar installed in Santa Cruz de Tenerife (Tenerife, Canary Islands,

Spain; 28° 30' N, 16° 12' W; 52 m a.s.l.) have been used to check the AOD stability. This instrument contains a solid-state laser system emitting at 532 nm in full-time continuous mode with a high-pulse repetition rate of 2500 Hz. More details of





this system and the on-site maintenance and calibration techniques are described by Campbell et al. (2002) and Welton and Campbell (2002).

## 2.3 ROLO model

RObotic Lunar Observatory (ROLO) model, developed as a part of the USGS and NASA-funded program for space-borne calibration, is considered an accurate tool for exo-atmospheric lunar spectral irradiance ($I_0$) estimation for a given position of the observer on the Earth's surface and at a given time (Kieffer and Stone, 2005). This empirically-based model provides Moon's irradiance at 32 wavelengths, with an uncertainty between 5% and 10% in the absolute scale, using only geometrical variables: the absolute phase angle, the selenographic latitude and longitude of the observer and the selenographic longitude of the Sun. $I_0$ values have been calculated in this work as the convolution of the product of Moon reflectances, calculated by Eq.

(10) in Kieffer and Stone (2005), the solar spectrum given by Wehrli (1986) and the Earth-Moon and Sun-Moon distances, with each of the CE318-T filter responses. The result must be multiplied by the solid angle of the Moon ($\sigma_M = 6.4177 \cdot 10^{-5}$ sr) divided by $\pi$, according to Kieffer and Stone (2005). The lunar ephemeris have been extracted using the Navigation Ancillary Information Facility (NAIF) of the NASA Jet Propulsion Laboratory (JPL) (http://naif.jpl.nasa.gov/naif.html), which uses data of the orbital position of many celestial bodies known as kernels or Spacecraft, Planet, Instrument, C-matrix (pointing), and

Events (SPICE) data files. This NAIF SPICE toolkit is free available at http://naif.jpl.nasa.gov/.

## 2.4 AOD retrieval method

Following Barreto et al. (2013a) and Barreto et al. (2016), AOD at night-time ($\tau_{a,night}$) for a given wavelength, $\lambda$, can be calculated using the following equation:

$$\tau_{a,night,\lambda} = \frac{\ln(\kappa_\lambda) - \ln(\frac{V_\lambda}{I_{0,\lambda}}) - m_{\text{atm}}(\theta) \cdot \tau_{\text{atm},\lambda}}{m_{\text{a}}(\theta)}. \tag{1}$$

In this expression $\kappa_\lambda$ is the calibration constant, $V_\lambda$ is the measured voltage, $I_{0,\lambda}$ is the extraterrestrial Moon irradiance given by the ROLO model, $m_{atm}$ and $\tau_{atm}$ are the air mass and the optical depth of all atmospheric attenuators with the exception of aerosols (Rayleigh scattering, and $O_3$ and $NO_2$ absorption), and $m_a$ is the aerosol air mass. $\theta$ stands for the Moon's zenith angle. Sub-index $\lambda$ makes reference to the respective $\lambda$-wavelength. All these terms have been calculated using the AERONET version 2 procedure (http://aeronet.gsfc.nasa.gov/new_web/data_description_AOD_V2.html).

Following the error propagation analysis performed by Barreto et al. (2016), the total combined uncertainty on the AOD retrieved using the CE318-T photometer is the sum of the relative uncertainties associated with the instrument calibration, the ROLO model ($\sim 1\%$, independent of orbital parameters) and instrumental errors. Only uncertainties related to instrumental errors were expected to be dependent on Moon's phase angle ($g$).





## 3 RESULTS

### 3.1 Identifying bias in the lunar irradiance model

A set of 59 nights characterized by pristine and stable AOD conditions at Izaña covering different Moon cycles, from March to December 2014, has been selected in this study. We have ensured stable AOD conditions using ancillary vertical information

from the MPL-3 lidar running at Santa Cruz de Tenerife station. These stable AOD conditions are confirmed by means of the range-corrected signal from the MPL-3 lidar at Santa Cruz and the AOD (day-time from AERONET and night-time from CE318-T, both at 500 nm) at Izaña. The MPL-3 profiles and the AOD evolution for one moon cycle in the period 3-17 October, 2014, are shown in Fig. 1. AOD is stable in the whole period with the exception of 3-5 October and 10 October, and these three nights are discarded from the fitting analysis. AERONET daytime AOD in this period ranges from 0.004 in 1640 nm to 0.016

in 440 nm.

Nocturnal measurements were performed by means of a master CE318-T installed in Izaña station. This instrument has been calibrated following the Lunar-Langley calibration method proposed by Barreto et al. (2013a). This is a new absolute calibration technique, specifically developed for lunar photometry, which is able to avoid the determination of one different calibration coefficient every night required by the common Langley technique (Shaw, 1976, 1983). In spite the simplicity of this

technique, its accuracy relies on the uncertainty involved in the ROLO model. The uncertainty estimation of the Lunar-Langley method was performed by Barreto et al. (2016) assuming a relative ROLO accuracy of $\sim 1\%$, independent of any orbital parameter. As a result, if the existence of some bias in the ROLO model is confirmed, the accuracy of the Lunar-Langley technique for absolute calibration should be revised.

CE318-T Lunar-Langley calibration was performed on three different nights within the 10 Moon-cycles period used in this

paper: 13 March, 12 June and 9 October. These three nights were characterized by Moon's illumination between 93% and 99%, with phase angles ($g$) between -31° and 19°. Meanwhile, CE318-T Langley calibration was performed on 51 different nights covering phase angles from -94° to 83°.

A comparison of the two absolute calibration techniques, Langley and Lunar-Langley, has been carried out in this paper in this 59 night-time period. In the case of $\kappa$'s obtained using the Lunar-Langley technique, we have calculated the average $I_0$

from the ROLO model during the calibration period to obtain the calibration coefficient ($V_0$). To this end, we have used the following equation:

$$V_{0,\lambda} = I_{0,\lambda} \cdot \kappa_\lambda \tag{2}$$

We found nearly stable $I_0$ values during the Langley period ($\leq 2h$), with average standard deviations below 1.6%. Relative $V_0$ differences with phase angle (Langley versus Lunar-Langley) are shown in Fig. 2 for 870 nm channel. Relative differences

$> 4\%$ are observed, especially near full Moon and near waning Moon. Small differences were found for phase angles between -20° and 60° ($<1\%$) and between 20° and 60° ($<2\%$), in addition to an asymmetry of the differences with phase angle (higher





differences after full Moon). It is worth mentioning the Lunar-Langley calibration technique systematically underestimates $V_0$ throughout the lunar cycle.

This phase angle dependence of the ROLO model has been also reported by Lacherade et al. (2013) and Viticchié et al. (2013), as well as its asymmetry within the Moon cycle (Lacherade et al., 2013).

5     It is important to highlight that Barreto et al. (2016) considered negligible the contribution of the covariance term in the combined uncertainty of two magnitudes expected to be correlated: $\kappa$ and $I_0$. This last assumption, which neglects the effect of possible Moon's irradiance uncertainties on calibration, is only valid considering that there are not relevant systematic errors in the irradiance model. Our results prove this statement is wrong, and the existence of a bias in the lunar irradiance model must be taken into account in lunar photometry.

10  **3.2   Correction of artificial AOD variations at night-time**

Once we have verified the existence of a bias in the lunar irradiance model which introduces calibration and AOD uncertainties dependent on Moon's phase angle, we propose an empirical correction method for the AOD retrieval at night-time. This method is exclusively focused on the AOD retrieval and it does not involve any correction to the lunar irradiance model.

We propose an empirical correction based on the use of day-time AOD information as proxy for nocturnal AOD providing 15  aerosol content remains stable. A total of 6997 night-time AOD measurements corresponding to the same 59 pristine and stable nights period (March to December 2014) in addition to 14575 daylight AOD measurements have been selected in this study. These clean conditions allow us to accurately estimate AOD at night-time considering a smooth AOD variation by linear interpolation using AERONET daylight information. AOD differences at night-time ($\Delta AOD_{fit}$) are defined through the comparison of the night-time AOD estimated from linear interpolation using AERONET daylight data ($AOD_{night,interp}$) and 20  the AOD obtained directly from nocturnal CE318-T measurements using Eq. (1) ($AOD_{night}$). Then, $\Delta AOD_{fit}$ can be obtained by means the following expression:

$$\Delta AOD_{fit} = \Delta AOD_{night,interp} - AOD_{night} \qquad (3)$$

These differences are displayed in Fig. 3a with asteriscs. We observe a clear dependence with the phase angle ($g$), increasing considerably for higher $g$ values. This dependence is more evident for higher wavelengths and seems somewhat asymmetric 25  (higher differences for positive $g$, namely, after full Moon, especially for 1020 nm channel), in agreement with the results obtained in Sect. 3.1. We have found the best fit for this dependency ($\delta_g$) to be a second-order polynomial (Eq. (4)), as it is displayed in Fig. 3a with solid lines.

$$\delta_g(\lambda) = a_0(\lambda) + a_1(\lambda) \cdot g + a_2(\lambda) \cdot g^2 \qquad (4)$$

The coefficients $a_i$ in Eq. (4) are calculated for each channel centered at $\lambda$. In addition, the presence of a nocturnal cycle 30  on $\Delta AOD_{fit}$ is also evident from Fig. 3a. These results are also in agreement with those found by Barreto et al. (2016), who



found higher AOD discrepancies with respect to daytime AERONET data for higher phase angles, and a faint nocturnal AOD cycle.

The increasing uncertainty in AOD with $g$ and the asymmetry of this dependence within the Moon cycle can be attributed, at least partially, to the existence of some residuals on the ROLO model, according to the results presented in Sect. 3.1. This

effect results in a systematic $g$-dependent AOD error observed in Fig. 3a, with values up to 0.035 before full Moon (1640 nm) and $> 0.06$ after full Moon (1020 nm), both for the highest phase angles. As Barreto et al. (2016) found, relative uncertainties associated with lunar measurements performed under higher phase angles (low illumination) are about 0.5% due to lower signal-to-noise ratio (SNR) compared with the 0.1% expected for near full Moon conditions. Note that these uncertainties change with $g$. However, this second factor introduces a random error, which we do not expect to introduce any bias in the

AOD retrieval process.

The observed AOD cycle at night-time is similar to the AOD cycle detected by many authors in sun photometry (Cachorro et al., 2004, 2008a, b), but with an amplitude dependent on $g$. The characteristics of this AOD cycle are expected to be similar to those of sun photometry: systematic and symmetrical around the lunar noon (Moon at maximum height), with maximum values at lunar noon and vanishing for larger air masses, being independent of AOD. The systematic artificial AOD behavior

and the persistence of this feature for different atmospheric conditions indicate it is not an atmospheric effect nor is it directly related to the ROLO model (ROLO's outputs are not $\theta$-dependent). It is likely produced by a calibration problem caused by the use of a biased ROLO model during calibration. Since the nocturnal calibration is normally performed at high illumination conditions, and AOD is subsequently calculated in these conditions using the ROLO irradiances ($I_0's$ in Eq. (1)), we suspect the possible bias in ROLO is cancelled for lower phase angles. However, when calibration constants (independent in essence

on Moon's phase) and the $I_0's$ are applied to low illumination conditions in AOD calculation, some $g$-dependent AOD residuals are expected to appear, causing the artificial nocturnal cycle observed in the AOD residuals. Since this systematic effect can be modulated by the inverse of air mass, we propose to include the effect of zenith angle on AOD difference as a function of $1/m_a$ ($\delta_\theta$ in Eq. (5)).

$$\delta_\theta(\lambda) = \frac{\beta_0(\lambda)}{m_a(\theta)} \qquad (5)$$

The coefficient $\beta_0$ in Eq. (5) is calculated for each channel centered at $\lambda$. The functional form of this corrective factor against the air mass (not including any dependence on g), shows that the closer the Moon is to the zenith the higher AOD differences are (Fig. 3b). We have displayed $\delta_\theta$ in a whole night at different phase angles and we found an impact of zenith angle on AOD between 0.005 (at 500 nm) and 0.016 (at 1020 nm), as it is shown in Fig. 4. A similar and symmetrical nocturnal cycle is displayed in the three nights, one near full moon ($g$ ranging from -9° to 3°) and two nights with low illumination conditions ($g$

between ±93° and ±96°), in which approximately the half of the Moon's trajectory between rising to setting times is displayed (due to sunlight).



### 3.3 Improvement of AOD retrieval at Izaňa

The two effects parameterized in Eqs. (4) and (5) are considered independent. Since these two variables are uncorrelated we propose a final parametrization based in these two effects: phase angle dependence (quadratic dependence) modulated by zenith angle. This parametrization, for each channel centered at $\lambda$, is presented in Eq. (6), where $\delta_{g,\theta}$ represents the functional form

for $\Delta\text{AOD}_{fit}$.

$$\delta_{g,\theta}(\lambda) = \frac{\beta_0(\lambda)}{m_a(\theta)} \cdot [a_0(\lambda) + a_1(\lambda) \cdot g + a_2(\lambda) \cdot g^2] \tag{6}$$

The main results for this regression analysis, including simultaneously $g$ and $\theta$ dependences, are presented in Table 1, in which we observe R-squared values ranging from 0.78 in 1020 nm to 0.60 in 440 nm. These results indicate this fitted model is able to account for $\sim 78\%$ and $60\%$ of the variation observed in $\Delta\text{AOD}_{fit}$ for 1020 nm and 1640 nm, respectively,

which are the channels more affected by the $g$ and $\theta$ dependence, within the 95% confidence bounds. RMSE values range from 0.009 in 1020 nm to 0.004 in 870 nm, indicating a low variance of the residuals. In fact longer wavelength channels show higher RMSEs because they are more sensitively affected by this dependence. In this respect we observe this empirical model is able to reproduce the asymmetry with phase angle, especially notable in case of 1020 nm. The total residual sum squares (SSE) of this fitting analysis are 0.19 for 1020 nm channel and below 0.10 for the rest of channels. This fitting

analysis can be used to define a corrected night-time AOD (AOD$_{corr}$) as the addition of the measured AOD (AOD$_{night}$) and $\delta_{g,\theta}$ (AOD$_{corr} = AOD_{night} + \delta_{g,\theta}$). The scatterplot between $\delta_{g,\theta}$ and $\Delta\text{AOD}_{fit}$ is shown in Fig. 5, where a good agreement between parameterized and measured differences is found in the case of shorter wavelengths ($\lambda \leq 870$ nm). The more important ones were retrieved for 1020 nm channel, and could be attributed to some temperature dependence at this spectral range.

We have performed an AOD day-night-day transition coherence test at Izaña in order to check this correction procedure.

In this test we have compared nocturnal (CE318-T master) and day-time (CE318-N AERONET master) AOD corresponding to the consecutive 1h time period during Moonset-Sunrise (MS-SR) and Sunset-Moonrise (SS-MR) in those 59 day-night-day transitions in 2014 classified as stable in terms of aerosol loads. In this test analysis $\Delta\text{AOD}_{trans}$ represents the AOD difference between 1h of sun and moon data. $\Delta\text{AOD}_{trans}$ before and after correction for phase angles ranging from -100° to 100° is presented in Fig. 6 and Table 2 for the six channels. We have verified the reduction in the systematic errors of

$\Delta\text{AOD}_{trans}$ after the correction, even though some problems of overcorrection are detected in the case of 1020 nm for $g > 50°$ and 500 nm for $g < -50°$. AOD correction near full Moon is very low, as we expected from Eq. (6). We have found AOD differences within $\pm 0.01$ after correction for any illumination condition, below the instrumental precision expected for the CE318-T photometer Barreto et al. (2016). This nocturnal AOD improvement is also evident in Fig. 7, especially for longer wavelength channels at low illumination conditions.

Finally, we present in Table 3 the mean AOD difference after and before the AOD correction ($\Delta\text{AOD}$) in the 59 nights AOD-stable period in 2014 at Izaña as a function of phase angle. These differences are below 0.01 in the case of near full moon conditions, but higher in the case of low illumination conditions: up to 0.04 and 0.03 for 1020 nm and 1640 nm, respectively,





and below 0.018 in the rest of channels. These results indicate the successful AOD correction, especially in longer wavelength channels, and the effective correction of the asymmetries through the Moon's cycle.

It is worth mentioning $\delta_{g,\theta}$ could be introduced in Eq. (1) to obtain $AOD_{corr}$, and therefore $\delta_{g,\theta}$ could be subsequently used to define an $I_0$ correction only dependent on Moon's phase angle ($I'_0 = I_0/\delta_g$).

## 3.4 Validation of the AOD correction at Izaña

We have carried out the validation of the AOD correction presented ($\delta_{g,\theta}$ in Eq. (6) with the coefficients in Table 1) using night-time AOD extracted from the CE318-T master at Izaña in the period June to August, 2016. This is a 43-night period which includes pristine conditions and some dust outbreaks with a maximum AOD at 500 nm of 0.61 (June, 2016). This period is suitable to assess the AOD correction in a time period different than that used to parameterize the correction empirical model (parameterization period) and at different AOD loads. Similar results were observed in this case (see Table 4), with average AOD differences (after and before correction) up to 0.037 (1020 nm) and 0.023 (1640 nm) in the case of low illumination conditions, and below 0.016 in the rest of channels. We have observed again, as in the parameterization period in 2014, a considerably reduction in the $g$ and $\theta$ dependence on AOD after correction.

### 3.5 Validation at other sites: Carpentras, Dakar and Lille

As a complementary validation analysis, we have extended the implementation of the previous correction procedure to other three test sites, affected by different aerosol conditions. We have applied to the nocturnal AOD obtained at these other sites the same AOD correction ($\delta_{g,\theta}$ in Eq. (6)) with the same coefficients presented in Table 1, in order to ensure the validity of this empirical model to sites with quite different aerosols content. Carpentras (44° 4' N, 5° 3' E, 100 m a.s.l.) and Lille (50° 36' N, 3° 8' E, 60 m a.s.l.), are two calibration sites located in France. Both sites are affected by relatively clean conditions and fine mode particles as the dominant aerosol size distribution (average AOD at 500 nm of 0.14 and 0.19, respectively, and average Angström Exponent of 1.4 and 1.2, respectively). In contrast, the station located in Dakar (M'Bour, Senegal, 14° 23' N, 16° 57' W, 0 m a.s.l.) presents a significant contribution of marine aerosols and biomass burning aerosols during the dry season, but also mineral dust with maximum influence in summer (Leon et al., 2009). In this case the averaged AOD at 500 nm and Angström Exponent are 0.45 and 0.37, respectively, indicating the predominant coarse-mode aerosol at this station.

We have used nocturnal CE318-T AOD data at Carpentras in a 9-night period from 15-24 February 2016, 10-night period at Dakar from 18-29 April 2016, and a 12-night period at Lille from 13-28 April 2016. These three instruments have been calibrated using the Lunar-Langley calibration technique at Izaña. AOD difference after and before correction ($\Delta$AOD) for these three stations are quantified in Table 4 and displayed for different phase angles in Fig. 8 (only for 1640 nm, 1020 nm and 440 nm, to improve visualization). We found $\Delta$AOD values similar to those obtained for Izaña, between 0.021 (1640 nm in Carpentras) and 0.005 (500 nm in Lille) in case of phase angle $g \leq$ -50° , below 0.01 in case of near full moon conditions, and ranging from 0.033 (1020 nm in Dakar) to 0.010 (675, 500 and 440 nm in Dakar) for $g \geq 50°$. It is worth mentioning we have observed at these three stations a similar asymmetry of $\Delta$AOD with the Moon's cycle previously detected at Izaña, in which AOD differences are higher for 1020 nm in case of high and positive phase angles (see Fig. 8b) while maximum differences





are observed in 1640 nm when phase angle $g$ is strongly negative (see Figs. 8a and 8c). These results corroborate the existence of a residual phase angle dependence on nocturnal AOD, and also exhibit a similar-in-magnitude zenith angle dependence as the one observed at Izaña, discarding the instrumental problem as the source of these errors.

## 4 CONCLUSIONS

The comparison of the CE318-T calibration performed by means of the Lunar-Langley and the calibration performed every single night by means of the common Langley technique indicates the existence of some systematic errors on the ROLO's lunar irradiances. These systematic errors could have an important impact on the AOD retrieved by means of lunar photometry. In order to reduce the uncertainties in the AOD retrieved at night-time using a calibration technique dependent on this lunar irradiance model (the Lunar-Langley and the intercomparison techniques Ratio Sun and Ratio Moon), we have studied the evolution of the AOD at night-time at Izaña high mountain observatory in a period characterized as clean and stable. These conditions were ensured by means of vertical profiles from an MPL-3 lidar installed in a nearby coastal station. We detected an important bias correlated to Moon's phase and zenith angles ($\theta$ and $g$), especially at longer wavelength channels. Working under stable AOD conditions, we have parameterized this residual dependence in nocturnal AOD in terms of Moon's phase and zenith angles through an empirical regression model. Our results show AOD at night-time is significantly corrected, with absolute errors $< 0.01$, i.e., below the instrument precision, in spite of the absence of a robust cloud screening system.

We attribute the phase angle dependence on AOD residuals to the inherent limitations of the ROLO model. The nocturnal cycle observed in AOD with a g-dependent amplitude could be related to the existence of a propagation of systematic calibration errors as a result of the use of a biased irradiance model for calibration (performed under high illumination conditions) and its subsequent use for AOD calculation. As a result, the use of a biased model during the Lunar-Langley calibration unavoidably introduces systematic calibration uncertainties that should be corrected using the empirical equations proposed in this study. The authors would like to admit this is only a preliminary AOD correction proposal which might be used to correct the lunar irradiance model. Since long-term lunar observations are required for an accurately modelling of the Moon's phase and librations effects, several years of lunar measurements are required to perform an adequate correction, in terms of both AOD or $I_0$. In this respect, we strongly recommend the use of the unique calibration method independent on any lunar irradiance model: the Sun-Moon Gain factor method, proposed by Barreto et al. (2016). Further investigations will be carried out to check the suitability of this technique for different locations and Moon's illumination conditions.

It should be highlighted that in this study the authors only intend to correct as much as possible the AOD retrieval at night-time affected by these biases. It is out of the scope of the present study to propose and perform corrections on the ROLO USGS model. This issue is beginning to be addressed thoroughly by several groups. An example is the Global Space-based Inter-Calibration System (GSICS) Lunar Observation Dataset (GLOD), a collaborative effort to use the Moon as a common and unique calibration reference at international level. This database includes Moon observations from several space organizations such as the European Space Agency (ESA), the European Organization for the Exploitation of Meteorological Satellites (EU-METSAT), the National Centre for Space Studies (CNES), The Japan Aerospace Exploration Agency (JAXA), the National



Satellite Meteorological Center from China Meteorological Administration (NSMC/CMA) or the National Aeronautics and Space Administration (NASA), among others.

*Acknowledgements.* This work has been developed within the framework of the activities of the World Meteorological Organization (WMO) Commission for Instruments and Methods of Observations (CIMO) Izaña Testbed for Aerosols and Water Vapor Remote Sensing Instruments.

5    AERONET Sun photometers at Izaña have been calibrated within the AERONET Europe TNA, supported by the European Union's Horizon 2020 research and innovation programme under grant agreement No 654109 (ACTRIS-2). The authors also thank AERONET team for their support. The GAW-PFR network for AOD at WMO-GAW global observatories has been implemented by the World Optical Depth Research and Calibration Center (WORCC).



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





**Table 1.** Results of $\Delta AOD_{fit}$ parameterization for each channel: model coefficients, coefficient of determination (R-squared) and root-mean-squared-error (RSME) within 95% confidence bounds.

| Channel (nm) | $a_0$ | $a_1$ | $a_2$ | $a_3$ | R-squared | RMSE |
|---|---|---|---|---|---|---|
| 1640 | 0.1260 | 0.0441 | 0.0040 | 0.0017 | 0.64 | 0.007 |
| 1020 | -0.7311 | -0.8115 | -0.0220 | -0.0004 | 0.78 | 0.009 |
| 870 | -0.0917 | -0.0919 | -0.0062 | -0.0007 | 0.78 | 0.004 |
| 675 | -0.0260 | -0.0132 | -0.0174 | -0.0002 | -0.71 | 0.005 |
| 500 | -0.0022 | -0.0001 | -0.0001 | -0.9755 | 0.68 | 0.005 |
| 440 | 0.0097 | 0.0155 | 0.0009 | 0.0037 | 0.60 | 0.006 |

**Table 2.** Averaged AOD differences between CE318-AERONET day-time and CE318-T night-time data ($\Delta AOD_{trans}$) during Sunset-Moonrise and Moonset-Sunrise in three ranges of Moon's phase angles ($g$) with and without $\delta_{g,\theta}$ correction.

| g range | # cases | correction | 1640 | 1020 | 870 | 675 | 500 | 440 |
|---|---|---|---|---|---|---|---|---|
| $\leq -50°$ | 39 | no | 0.024 | 0.019 | 0.017 | 0.014 | 0.011 | 0.010 |
| | | yes | 0.008 | 0.007 | 0.007 | -0.003 | -0.005 | -0.003 |
| $-50° > g > 50°$ | 81 | no | 0.011 | -0.003 | 0.008 | 0.007 | 0.005 | 0.002 |
| | | yes | 0.008 | -0.001 | 0.006 | 0.005 | 0.004 | 0.002 |
| $\geq 50°$ | 26 | no | 0.022 | 0.025 | 0.014 | 0.013 | 0.010 | 0.010 |
| | | yes | 0.002 | -0.006 | 0.001 | 0.002 | -0.001 | 0.001 |

**Table 3.** Average AOD difference between corrected and non-corrected AOD ($\Delta AOD$) measured at Izaña station as a function of the Moon's phase angle ($g$) in degrees in a 59 nights period transition coherence test in 2014. Number of night-time measurements ($N$) is included.

| g range | $N$ | 1640 | 1020 | 870 | 675 | 500 | 440 |
|---|---|---|---|---|---|---|---|
| $\leq -50°$ | 891 | 0.020 | 0.015 | 0.012 | 0.014 | 0.008 | 0.008 |
| $-50° > g > 50°$ | 5076 | 0.003 | 0.003 | 0.002 | 0.002 | 0.001 | 0.001 |
| $\geq 50°$ | 1030 | 0.026 | 0.041 | 0.018 | 0.014 | 0.014 | 0.013 |



**Table 4.** Average AOD difference between corrected and non-corrected AOD ($\Delta$AOD) measured at four different stations in 2016. Number of night-time measurements ($N$) are included.

| g range | station | $N$ | 1640 | 1020 | 870 | 675 | 500 | 440 |
|---|---|---|---|---|---|---|---|---|
| $\leq$ -50° | Izana | 870 | 0.016 | 0.011 | 0.010 | 0.012 | 0.006 | 0.006 |
| | Carpentras | 350 | 0.021 | 0.015 | 0.013 | 0.015 | 0.008 | 0.009 |
| | Lille | 55 | 0.015 | 0.009 | 0.010 | 0.011 | 0.005 | 0.006 |
| | Dakar | 0 | — | — | — | — | — | — |
| -50° > g > 50° | Izana | 3459 | 0.003 | 0.004 | 0.002 | 0.002 | 0.001 | 0.001 |
| | Carpentras | 660 | 0.003 | 0.001 | 0.002 | 0.002 | 0.001 | 0.001 |
| | Lille | 423 | 0.003 | 0.001 | 0.002 | 0.002 | 0.001 | 0.001 |
| | Dakar | 527 | 0.002 | 0.002 | 0.002 | 0.002 | 0.001 | 0.001 |
| $\geq$ 50° | Izana | 631 | 0.023 | 0.037 | 0.016 | 0.012 | 0.012 | 0.011 |
| | Carpentras | 0 | — | — | — | — | — | — |
| | Lille | 7 | 0.011 | 0.018 | 0.008 | 0.006 | 0.006 | 0.005 |
| | Dakar | 31 | 0.020 | 0.033 | 0.014 | 0.010 | 0.010 | 0.010 |





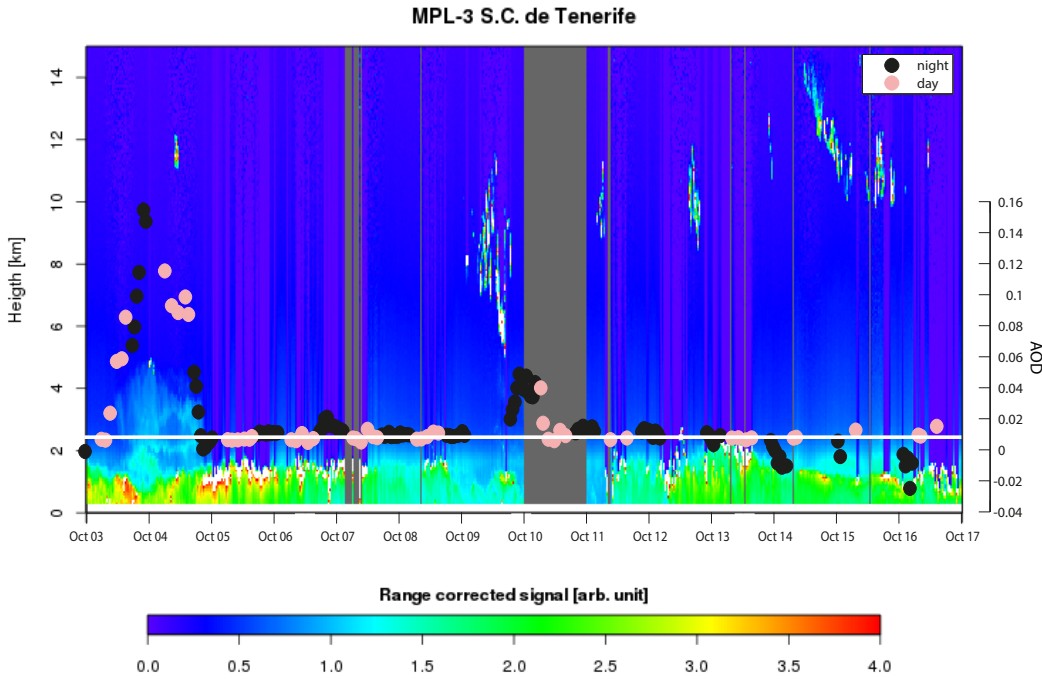

**Figure 1.** Lidar range-corrected backscattering signal from the MPL-3 installed at Santa Cruz and AOD (at 500 nm) evolution (day in pink circles and night in black circles) extracted from the CE318-T installed in Izaña in October, 2014. Horizontal white line represents the Izaña's Observatory altitude.



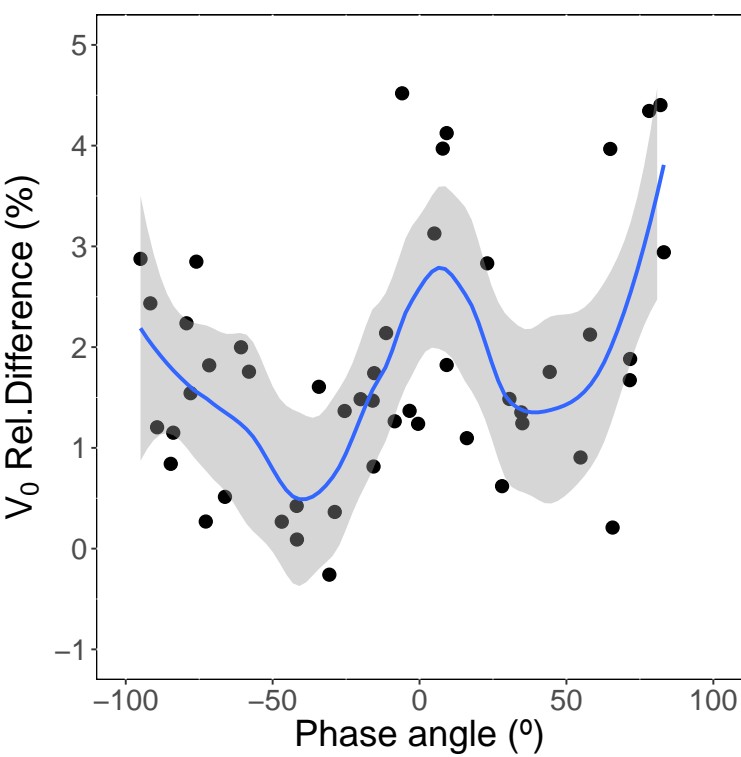

**Figure 2.** Calibration coefficients ($V_0$) relative difference (%) at 870 nm when Langley and Lunar-Langley absolute calibration techniques are compared. Smoothing by means of LOESS (locally estimated scatterplot smoothing) is shown with solid line. The shaded areas represent the 95% confidence interval.

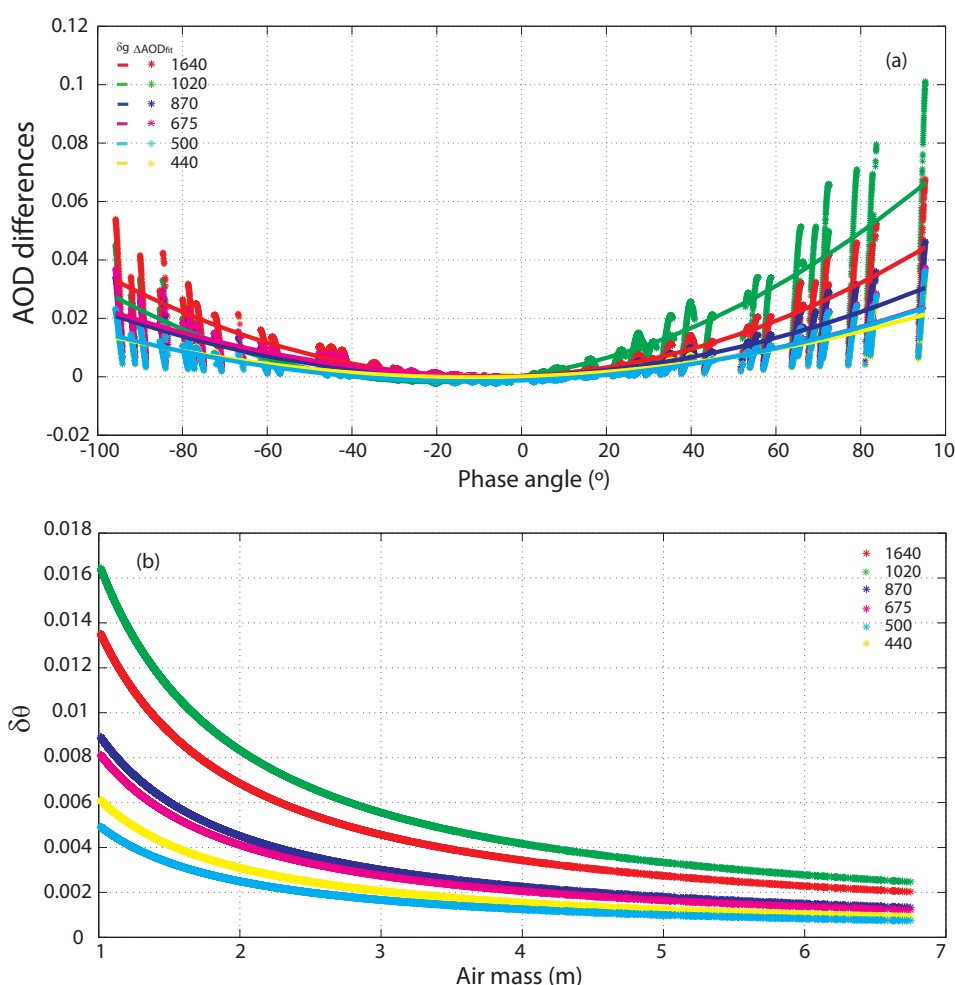

**Figure 3. (a)** AOD differences between daylight interpolated and night-time measured values ($\Delta AOD_{fit}$ with data points) and AOD differences predicted from the fitting analysis with Moon's phase angle ($\delta_g$ with coloured lines). **(b)** AOD differences predicted from the fitting analysis with Moon's zenith angle ($\delta_\theta$).





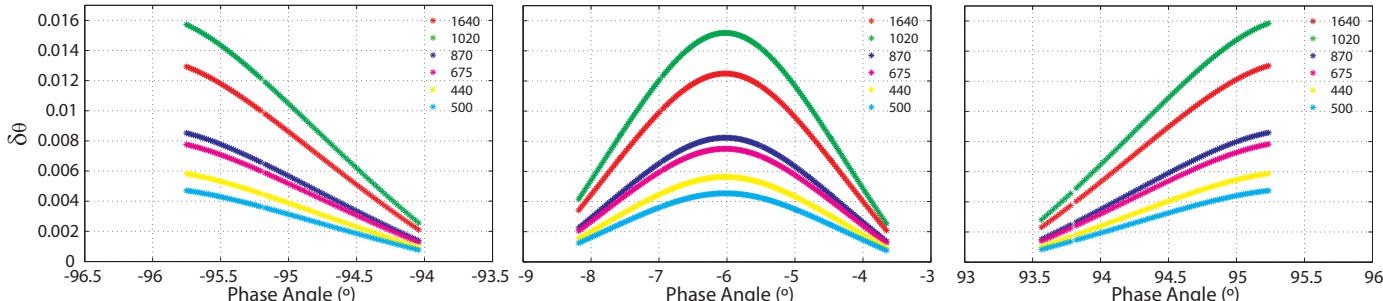

**Figure 4.** Predicted AOD differences from the fitting analysis with $\theta$ ($\delta_\theta$) for different phase angles.

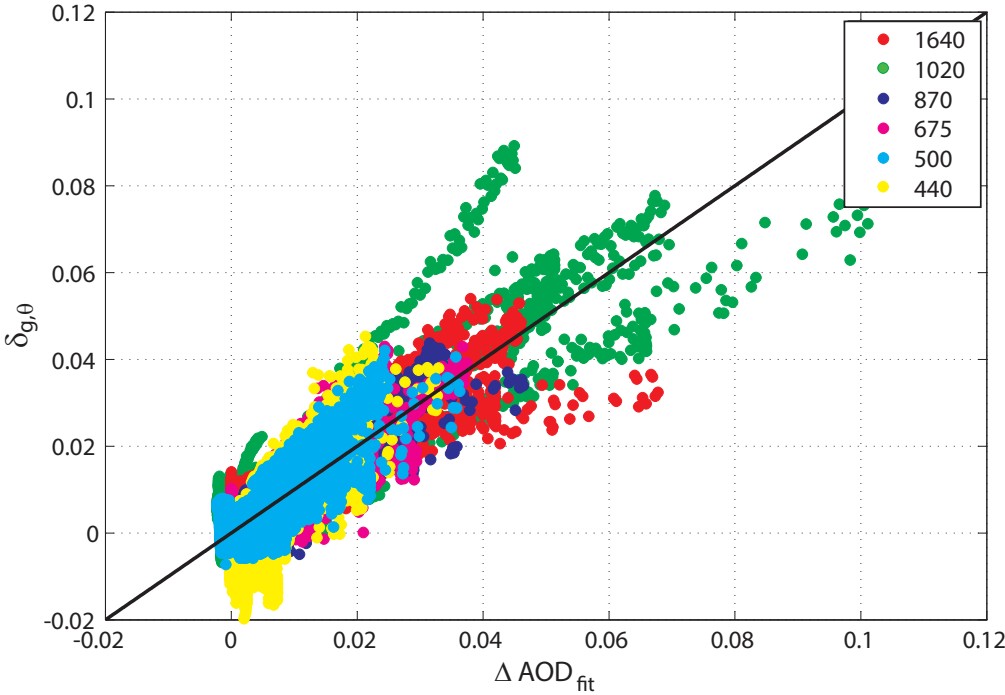

**Figure 5.** Scatterplot with parameterized ($\delta_{g,\theta}$) and measured ($\Delta \mathrm{AOD}_{fit}$) AOD differences. Solid line is the diagonal ($y = x$).





**Figure 6.** AOD differences with the Moon's phase angle in the AOD day/night transition coherence test between day-time CE318-N AERONET AOD and CE318-T night-time AOD during Sunset-Moonrise and Moonset-Sunrise period before (circles) and after (triangles) the AOD correction.







**Figure 7.** AOD evolution for one Moon cycle (June, 2014) at Izaña. Opaque colors represent daylight data. Nocturnal AOD before **(a)** and after **(b)** correction is plotted.







**Figure 8.** AOD before correction (BC) and after correction (AC) on AOD in **(a)** Carpentras, **(b)** Dakar and **(c)** Lille in 2016. Opaque colors represent daylight data. The black line and right y axis correspond to the phase angle evolution in this period.