# Peer review of "Assessment of nocturnal Aerosol Optical Depth from lunar photometry at Izaña high mountain Observatory"

_Atmospheric Measurement Techniques, 2016_

## Referee Comment (RC1) · Anonymous Referee #1 · 27 Mar 2017

This manuscript by Barreto et al., "Assessment of nocturnal Aerosol Optical Depth from lunar photometry at Izaña high mountain Observatory", describes the investigation and correction approach of errors affecting AOD calculation from Moon photometry. If find the manuscript interesting and relevant for the topic of Moon photometry. The topic is also suitable to the scope of AMT. The manuscript is well written and I would recommend it to be published after minor/technical corrections.

Minor comments:

1.

- P9.L17-18. What did you mean with "more important ones"? I think it refers to the

increased deviation of the 1020nm channel compared to other channels in Fig.5. It looks like you can split the data of the 1020nm channel in Fig.5 to three, or maybe four, different cases with different slopes. Could this be related to temperature differences, as it is stated in the same sentence?

2.

- P4.22-23. Another question to concerning the 1020nm channels. There are two channels with a nominal wavelength of 1020nm. Are both combined to one channel, or which channel measurements are used in this manuscript? Also it is stated, that the silicon 1020nm channel is temperature corrected. How could the deviation of the 1020nm channel in Fig.5 be related to the temperature than?(see comment 1)

Technical comments:

- P8.L29 "the three nights": Which three nights? Either spare the "the" or give the dates.

- P8.L29 "ranging from -9° to 3°" → ranging from -9° to -3°

- Fig4. I think it would be good to provide the dates.
* * *

---

## Referee Comment (RC2) · Anonymous Referee #2 · 13 Apr 2017

General

This manuscript presents an empirical approach to correct for systematic errors in AOD retrieved from lunar photometry. A simple parameterization based on moon phase and zenith angles is proposed to correct nocturnal AOD. By means of considering "stable conditions" daytime AOD was interpolated to obtain nighttime values which were compared with (corrected) AOD from lunar photometry. The correction appears to have a favourable effect. The correction procedure has been applied to three other stations to show the correction effect on AOD for different aerosol regimes.

The paper is well written and is relevant for publication in AMT. The AMT review criteria are all met, except # 13 ("Should any parts of the paper (text, formulae, figures, tables)

[Figure]

be clarified, reduced, combined, or eliminated?") which refers to the Introduction (which is a bit lengthy) and Figure 8 (which may be improved). I will explain these points below. In general, the paper is of good quality and I suggest only minor revisions.

Specific comments

- As stated above, the Introduction is rather long compared to the other sections of the paper. A possible solution may be to split up the introduction into a more concise version and a second section containing background information and references. In my view this will add to the readability of the paper.

- In Sections 2 and 3 the two methods for calibration are mentioned; the "common Langley technique" and the "Lunar-Langley technique". Although well-informed readers are familiar with the two techniques I suggest to spend a few more words on the techniques and its limitations, in particular the "every-night requirement" for the common technique (including an explanation).

- Section 3.1: it is stated that stable AOD conditions are selected using ancillary vertical information from an MPL-3 lidar. Although Fig. 1 is useful I suggest to give some more information on what exactly is meant by "stable" and how stable periods are selected (is there a quantitative criterion?). The interpolation method to get nighttime AOD from daytime values is a crucial aspect in the paper, so the background of how to select stable data deserves more attention/explanation.

- An interesting and good aspect of the work is that the AOD correction has been applied to three other stations in different aerosol climates. The question that arises is: is the correction (the coefficients presented in Table 1) fully instrument-independent? Since the correction is wavelength-dependent, the assumption that the parameterization can be applied to other instruments seems to rely on comparable spectral characteristics (in particular the filters) of the reference instrument and the instruments to which the correction is applied. I suggest the authors spend a few words on explaining why the correction can safely (?) be applied to the Carpentras, Dakar and Lille

instruments, and on the possible introduction of extra uncertainty.

- The title of Section 3.5 is not entirely correct since the method is not really validated. The authors have applied their correction method to AOD retrievals at the three sites but a real validation has not been performed. I suggest changing the title into something more appropriate ("Impact of correction. . ." or something equivalent). - Section 3.4 (line 19): Lille is considered to be affected by "relatively clean conditions". I think that this cannot be stated in general considering the highly polluted environment the site is located in. The AOD may be not particularly high during the selected period but "clean conditions" is maybe a bit too optimistic.

Technical corrections

- Table 1: R-squared: the value –0.71 seems to be incorrect.

- Although Fig. 8 can be understood, it is not very easy to distinguish between the asterisks and the circles. It maybe an idea to present AOD differences instead of absolute values in order to avoid unclearness. If this is not desirable, the authors may think of another way to make the figure more clear.

---

## Author Comment (AC3) · 12 Jun 2017

The authors want to highlight, regarding the Comment #2, that the two measurements at 1020 nm performed with the silicon and the InGaAs detectors are used as quality control to assess the stability of measurements performed with both detectors. However, only 1020 nm silicon measurements are commonly used to derive AOD.

---

## Author Response (AR1)

**Author's Response**

These are the corrections included in the manuscript:

**AUTHORS**

We have included a new coauthor: Dr. Carmen Guirado (Valladolid University, Izaña Atmospheric Research Center). She is in charge of the operation, calibration and quality assurance of the AERONET data in Izaña, which has served us to perform the AOD correction presented in this paper. So, we consider she has actively contributed to the present study.

We have also changed the order of the affiliations.

**INTRODUCTION**

According to Referee's 2 comments (C1), we have reduced the introduction to improve the readability of the paper. This is the new introduction:

[revised manuscript text omitted]

**SECTION 2**
Pag. 4, line 10: "… as well as two additional **measurements** at 1020 nm and 1640 nm using…"
Pag. 4, line 11: "The UV **spectral bands** do not …"

Pag. 5, line 9: The sentence has been slightly changed in order to highlight we are using an implementation of the ROLO model based exclusively on Eq. 10 published by Kieffer and Stone (2005): "…, calculated using our own implementation of the ROLO model based rigorously on the Eq. (10) published in Kieffer and Stone (2005)…"

Pag. 5, line 24: According to Referee's 2 comment (C2) we have included the following information about the Lunar Langley as well as the Langley methods.

*"…AOD_V2.html).*

*The calibration constant $\kappa_\lambda$ has been calculated by means the Lunar-Langley method developed by Barreto et al. (2013a). The main equations involved in this method are the Eq. 2, derived from the Beer-Lambert-Bouguer Law (the basis of the Langley calibration technique described by many authors in sunphotometry, such as Shaw, 1976, 1983, Holben, 1998, among others) and the Eq. 3 (the basis of the Lunar-Langley calibration technique), which defines the calibration constant as the ratio of $V_0$ to $I_0$.*

$$ln(V_\lambda) = ln(V_{0,\lambda}) - m(\theta) \cdot \tau_\lambda \qquad (2)$$

$$V_{0,\lambda} = I_{0,\lambda} \cdot \kappa_\lambda \qquad (3)"$$

**SECTION 3**
Pag. 6, line 7. As Referee 2 suggested (C3), we have completed the information about how stable AOD conditions are verified: "… 500 nm) at Izaña. **The AOD stability criterion involves an AOD difference between the 1-h average AERONET AOD of two consecutives days (sunset versus sunrise) ≤ 0.005 at 870 nm.**".

Pag. 6, line 11. Regarding the previous Referee's 2 comment (C2) we have included the following information: *"Nocturnal measurements were performed by means of a master CE318-T installed at Izaña station. This instrument has been calibrated following the Lunar-Langley calibration method proposed by Barreto et al. (2013a)* **(Eqs. 2 and 3.** *This is a new absolute calibration technique, specifically developed for lunar photometry, which avoids the determination of one different calibration coefficient every night required by the common Langley technique* **(Eq. 2)**. *It is important to emphasize the moon's illumination variation inherent to the lunar cycle, which means the $V_0$ and $I_0$ terms in Eq. 3 are continuously changing, even during the ≈ 2 hour observation time period required to perform the Langley calibration. Even if we discard the $I_0$ variation during the Langley period, the extraterrestrial voltages $V_0$ should be determined every day, which is not plausible considering the restrictive requirements in terms of atmospheric stability and cloudiness of this calibration technique. In spite the simplicity of the Lunar-Langley technique, its accuracy relies on the uncertainty involved in the ROLO model. The uncertainty …"*

Pag. 6, line 23: "A comparison of the two absolute calibration techniques, Langley and Lunar-Langley, has been carried out in this paper."

Pag 6, line 26: "To this end we have used the **Eq. 3**."

Pag. 7, line 3-4: Added one reference as well as the following information: "This phase angle dependence of the ROLO model has been also reported by Lacherade et al. (2013, **2014**) and Vitichie et al. (2013), as well as its asymmetry within the Moon cycle Lacherade et al. (2013, **2014**). **Lacherade et al. (2014) found a variation up to 5% with the phase angle between ±90°, the validity range of the ROLO model. These results are in agreement with the relative differences higher than 4% found in this study.**".

Pag. 8, line 29. We have included the dates for these three nights (see Referee 1 comment #3). We have included: "… one near full moon (**8 October,** with g ranging between -9° to -3°) and two nights with low illumination conditions (**6 April and 16 October**, with g between ±93° and ±96°), in which…".

Pag. 8, line 29. We have also corrected the typo error (see Referee 1 comment C4): "…with g ranging between -9° to **-3°**)".

Pag. 9, line 10: "… which are the **spectral bands** more affected…."

Pag. 9, line 11: "… longer wavelength **bands** show…."

Pag. 9, line 14: "… are 0.19 for 1020 nm and below 0.10 for the rest of **bands**…."

Pag. 9, line 18. According to Referee's 1 (comment #1), we have introduced new clarifications about the different slopes observed in Fig. 5 in the case of 1020 nm spectral band. We have included the following information:
"**The more important differences were retrieved for 1020 nm spectral band. We attribute the two branches above and below the horizontal line to a systematic error in our empirical model, which reproduces an amplified phase angle dependence in this 1020 nm spectral band. This effect is less appreciable but still discernible for 1640 nm. The points above the diagonal correspond to overcorrected AOD values. It happens for high and positive phase angles. On the contrary, the points below the diagonal line represent those conditions poorly corrected, and happens for high and negative phase angles. Finally, the third branch with $\delta_{g,\theta}$ values up to 0.09 is observed for high and positive phase angles in some days in October and November, 2014. We suspect that instrumental problems are behind such overcorrection cases.**"

Pag. 9, lines 21 and 23: We have changed "1h" for "1-h".

Pag. 9, line 23: We have changed "sun and moon" for "Sun and Moon".

Pag. 10, line 5. The title in section 3.4 has been changed by this one "Evaluation of the AOD correction at Izaña".

Pag. 10, line 12: "… in the rest of **spectral bands**."

Pag. 10, line 14. The title in section 3.5 has been changed by this one "Evaluation of the AOD correction at other sites: Carpentras, Dakar and Lille", according to Referee's 2 (C5).

Pag. 10, line 19. Changed "relatively clean conditions" for "relatively low AOD conditions".

Pag. 11, line 3: "This evaluation analysis in different stations seems to corroborate that this correction procedure is applicable to other instruments and sites. However, it is fair to admit this correction has been performed by means of an unique instrument, with certain optical interference filters. The difference in the filter responses as well as the degradation of optical filters with time are the limiting factors. They could add an extra uncertainty depending on the different band responses between instruments. Further studies will be focused on the estimation of this extra uncertainty."

Pag. 11, line 11: "… in a nearby coastal station **as well as by means of an AOD stability criterion using daytime AERONET data**.".

Pag. 11, line 12: "We detected an important bias correlated to Moon's phase and zenith angles **(g and θ) in all the spectral bands. However, the important phase angle dependence found for 1020 nm and 1640 nm might be an artifact caused by an systematic error in our empirical model**."

Pag. 11, line 21: "The authors would like to admit this is only a preliminary AOD correction proposal **developed using one single instrument** which might be **refined** and used to correct the lunar irradiance model **in future studies**."

**TABLES**
Table 1: Corrected the typo error (-0.71 for 0.71). See Referee's 2 comment #7.

**FIGURES**
Fig. 4: We have added information about the dates of these three nights, according to Referee's 1 comment #5.

Fig. 8: We have improved this figure to address the Referee's 2 comment #8. We have enlarged this figure as well as decreased the frequency of points in the case of Carpentras.

**Response to Referee #1**

The authors would like to thank the referee #1 for their constructive and useful suggestions. Please, find below our answers to their comments.

>> C1 P9.L17-18. What did you mean with "more important ones"? I think it refers to the increased deviation of the 1020nm channel compared to other channels in Fig.5. It looks like you can split the data of the 1020nm channel in Fig.5 to three, or maybe four, different cases with different slopes. Could this be related to temperature differences, as it is stated in the same sentence?

With "the more important ones" we are referring to the highest $\Delta AOD_{fit}$ values. We agree these different slopes should be clarified. Even though it is well known and documented the temperature dependence of 1020nm measurements performed with the common silicon detectors, we have to admit that the three branches observed in Fig. 5 are not caused by this effect, but by the strong asymmetry with phase angle found in the empirical model parameterized as $\delta_{g,\theta}$ for 1020nm and 1640nm spectral bands. So, we have two characteristic branches in these two bands: one above and one below the diagonal line. The points above the diagonal line correspond to overcorrected AOD values. It happens for high positive phase angles. On the contrary, the points below the diagonal represent those conditions poorly corrected, and it happens for high negative phase angles. We consider this is a systematic error in our empirical model which reproduces an amplified phase angle dependence in these spectral bands.

Finally, the first and most obvious of branches , with $\delta_{g,\theta}$ values up to 0.09, is observed in the case of high and positive phase angles in some days in October and November, 2014. We suspect that instrumental problems are behind such overcorrection cases.

We will introduce in the text these new clarifications, highlighting that the important phase angle dependence found in 1020nm and 1640nm channels might be an artifact.

>> C2 P4.22-23. Another question to concerning the 1020nm channels. There are two channels with a nominal wavelength of 1020nm. Are both combined to one channel, or which channel measurements are used in this manuscript? Also it is stated, that the silicon 1020nm channel is temperature corrected. How could the deviation of the 1020nm channel in Fig.5 be related to the temperature then? (see comment 1).

Yes, the CE318-T has one spectral filter with 1020nm as nominal wavelength, and two silicon and InGaAs detectors can be additionally used to measure in this spectral band. The one used in this paper is the silicon 1020nm channel. See comment 1.

>> C3 P8.L29 "the three nights": Which three nights? Either spare the "the" or give the Dates.

We will add the dates of these three nights (April 6$^{th}$, October 8$^{th}$ and October 16$^{th}$).

>> C4 P8.L29 "ranging from -9° to 3°" → ranging from -9° to -3°

Done.

>> C5 Fig4. I think it would be good to provide the dates.
We will provide dates (see comment 3).

**Response to Referee #2**

We thank the reviewer #2 for the positive and constructive comments. Listed below are our responses to the eight different discussion points.

>> C1 As stated above, the Introduction is rather long compared to the other sections of the paper. A possible solution may be to split up the introduction into a more concise version and a second section containing background information and references. In my view this will add to the readability of the paper

According to the referee's comment we have reduced the introduction in order to improve the readability of the paper.  This is the new introduction:

[revised manuscript text omitted]

>> C2 - In Sections 2 and 3 the two methods for calibration are mentioned; the "common Langley technique" and the "Lunar-Langley technique". Although well-informed readers are familiar with the two techniques I suggest to spend a few more words on the techniques and its limitations, in particular the "every-night requirement" for the common technique (including an explanation).

Information about the common Langley and the Lunar Langley as well as their limitations has been included in the text as follows:

Pag. 5, line 24: *"…AOD_V2.html). The calibration $\kappa_\lambda$ is calculated by means the Lunar-Langley method developed by Barreto et al. (2013a). The main equations involved in this method are the Eq. 2, derived from the Beer-Lambert-Bouguer Law (the basis of the Langley calibration technique described by many authors in sunphotometry -Shaw, 1976, 1983, Holben, 1998; among others-) and the Eq. 3 (the basis of the Lunar-Langley calibration technique), which defines the calibration constant as the ratio of $V_0$ to $I_0$.*

$$ln(V_\lambda) = ln(V_{0,\lambda}) - m(\theta) \cdot \tau_\lambda \tag{2}$$

$$V_{0,\lambda} = I_{0,\lambda} \cdot \kappa_\lambda \tag{3}"$$

Pag. 6, line 11: *"Nocturnal measurements were performed by means of a master CE318-T installed at Izaña station. This instrument has been calibrated following the Lunar-Langley calibration method (Barreto et al., 2013a) previously described. This is a new absolute calibration technique, specifically developed for lunar photometry, which avoids the determination of one different calibration coefficient every night required by the common Langley technique (Shaw, 1976, 1983). It is important to emphasize the moon's illumination variation inherent to the lunar cycle, which means the $V_0$ and $I_0$ terms in Eq. 3 are continuously changing, even during the ≈ 2 hour observation time period required to perform the Langley calibration. Even if we discard the $I_0$ variation during the Langley period, the extraterrestrial voltages $V_0$ should be determined every day, which is not plausible considering the restrictive requirements in terms of atmospheric stability and cloudiness of this calibration technique. In spite the simplicity of the Lunar-Langley technique, its accuracy relies on the uncertainty involved in the ROLO model."*

>> C3 Section 3.1: it is stated that stable AOD conditions are selected using ancillary vertical information from an MPL-3 lidar. Although Fig. 1 is useful I suggest to give some more information on what exactly is meant by "stable" and how stable periods are selected (is there a quantitative criterion?). The interpolation method to get nighttime AOD from daytime values is a crucial aspect in the paper, so the background of how to select stable data deserves more attention/explanation.

We have verified the stable AOD conditions with MPL-3 lidar aerosol backscattering information, and using an AOD stability criterion in the 1-h average AERONET AOD of two consecutive days at 870 nm. Doing so, the night is considered stable if the average of the AOD difference between two consecutive days is ≤ 0.005. This information will be included in the final manuscript.

>> C4 An interesting and good aspect of the work is that the AOD correction has been applied to three other stations in different aerosol climates. The question that arises is: is the correction (the coefficients presented in Table 1 fully instrument-independent?

Since the correction is wavelength-dependent, the assumption that the parameterization can be applied to other instruments seems to rely on comparable spectral characteristics (in particular the filters) of the reference instrument and the instruments to which the correction is applied. I suggest the authors spend a few words on explaining why the correction can safely (?) be applied to the Carpentras, Dakar and Lille instruments, and on the possible introduction of extra uncertainty.

Our results showed that the correction can be applied to any other instrument and location. However, we will include in the text the following information:

Pag. 11, line 3: "This evaluation analysis in different stations seems to corroborate the applicability of this correction procedure to other instruments and sites. However, it is fair to admit this correction has been performed by means of an unique instrument, with certain optical interference filters. The difference in the filter responses as well as the degradation of optical filters with time are the limiting factors. They could add an extra uncertainty depending on the different band responses between instruments. Further studies will be focused on the estimation of this extra uncertainty."

>> C5 The title of Section 3.5 is not entirely correct since the method is not really validated. The authors have applied their correction method to AOD retrievals at the three sites but a real validation has not been performed. I suggest changing the title into something more appropriate ("Impact of correction. . ." or something equivalent).
We agree with this comment. We will replace the current title by "Evaluation of the AOD empirical correction at other sites: Carpentras, Dakar and Lille".

>> C6 Section 3.4 (line 19): Lille is considered to be affected by "relatively clean conditions". I think that this cannot be stated in general considering the highly polluted environment the site is located in. The AOD may be not particularly high during the selected period but "clean conditions" is maybe a bit too optimistic
We agree. We will replace "clean" by "relatively low AOD conditions".

>> C7 Table 1: R-squared: the value –0.71 seems to be incorrect.
We have corrected this mistake.

>> C8 Although Fig. 8 can be understood, it is not very easy to distinguish between the asterisks and the circles. It maybe an idea to present AOD differences instead of absolute values in order to avoid unclearness. If this is not desirable, the authors may think of another way to make the figure more clear.
AOD differences don't show whether the correction actually improves the AOD retrievals, so we consider AOD absolute values the best way to confirm the correction presented in this work functions adequately. However, we have improved this figure to make it clearer to the reader. We have enlarged the figure and we have also decreased the frequency of points in the case of Carpentras site.

[revised manuscript text omitted]

---

## Author Response (AR2)

**Author's Response**

These are the corrections included in the manuscript:

- p. 5, l. 10: chemical symbols $O_3$ and $NO_2$ have been changed to normal font

- p. 7, Eq. 4: \Delta AOD (as acronym) has been changed to normal font.

- p. 9, l. 17/18: ... the 1020 spectral band ...

- p. 11, l. 8: in different > at different

[revised manuscript text omitted]